# Changes in Phenotypic and Molecular Features of Naïve and Central Memory T Helper Cell Subsets following SARS-CoV-2 Vaccination

**DOI:** 10.3390/vaccines12091040

**Published:** 2024-09-11

**Authors:** Mia Mosavie, Jennifer Rynne, Matthew Fish, Peter Smith, Aislinn Jennings, Shivani Singh, Jonathan Millar, Heli Harvala, Ana Mora, Fotini Kaloyirou, Alexandra Griffiths, Valerie Hopkins, Charlotte Washington, Lise J. Estcourt, David Roberts, Manu Shankar-Hari

**Affiliations:** 1Centre for Inflammation Research, Institute of Regeneration and Repair, University of Edinburgh, 4–5 Little France Drive, Edinburgh EH16 4UU, UK; 2Center for Bacterial Pathogenesis, Division of Infectious Diseases, Massachusetts General Hospital, Boston, MA 02114, USA; 3Department of Medicine, Charing Cross Hospital, Imperial College Healthcare NHS Trust, London W6 8RF, UK; 4Radcliffe Department of Medicine, University of Oxford, Oxford OX3 9DU, UK; 5Microbiology Services, Colindale, NHS Blood and Transplant, Colindale NW9 5BG, UK; 6Heart Lung Research Institute Clinical Research Facility, Cambridge CB2 0BB, UK; 7Statistics and Clinical Research, NHS Blood and Transplant, Cambridge CB2 0PT, UK; 8Statistics and Clinical Research, NHS Blood and Transplant, Bristol BS34 7QH, UK; 9Donor Medicine, NHS Blood and Transplant, Birmingham B15 2SG, UK; 10Department of Critical Care Medicine, Royal Infirmary of Edinburgh, Edinburgh EH16 4SA, UK

**Keywords:** SARS-CoV-2, COVID-19, vaccination, CD4^+^ T helper cell, transcriptomics, epigenetics, repertoire, T cell receptor

## Abstract

Molecular changes in lymphocytes following SARS-CoV-2 vaccination are incompletely understood. We hypothesized that studying the molecular (transcriptomic, epigenetic, and T cell receptor (TCR) repertoire) changes in CD4^+^ T cells following SARS-CoV-2 vaccination could inform protective mechanisms and refinement of future vaccines. We tested this hypothesis by reporting alterations in CD4^+^ T cell subsets and molecular features of CD4^+^ naïve and CD4^+^ central memory (CM) subsets between the unvaccinated and vaccinated groups. Compared with the unvaccinated, the vaccinated had higher HLA-DR expression in CD4^+^ T subsets, a greater number of differentially expressed genes (DEGs) that overlapped with key differentially accessible regions (DARs) along the chromatin linked to inflammasome activation, translation, regulation (of apoptosis, inflammation), and significant changes in clonal architecture beyond SARS-CoV-2 specificity. Several of these differences were more pronounced in the CD4^+^CM subset. Taken together, our observations imply that the COVID-19 vaccine exerts its protective effects via modulation of acute inflammation to SARS-CoV-2 challenge.

## 1. Introduction

Severe acute respiratory syndrome coronavirus 2 (SARS-CoV-2) vaccines to safely protect patients from developing more severe coronavirus disease 2019 (COVID-19) were authorized for use within a year of publication of the viral sequence [1]. The mRNA vaccine using lipid nanoparticle application [2,3] and the adenovirus vector-based vaccine containing a DNA vector to encode the spike protein [4,5] induce protective immune responses via mRNA translation [5].

Immune responses following SARS-CoV-2 vaccination regimes highlight higher frequencies of CD4^+^ helper T cells producing effector cytokines [6] and a sustained anti-viral response for at least 6–15 months [7]. The immunological effects, including duration of anti-viral responses, are higher with two-dose vaccinated individuals compared with a single-dose. Specifically for our hypothesis, the beneficial effects of a two-dose vaccination schedule have been linked to decreases in the “epigenetic age” in older individuals [8]. Importantly, the DNA methylation profiles (referred to as “epigenetic age”) have also been linked to the severity of COVID-19 [9]. Thus, we hypothesized that understanding molecular (transcriptomic, epigenetic landscape, and repertoire levels) changes following vaccination could provide insights on the biological basis for these protective immune responses and enable refinement of future vaccines and that the vaccine effects on naïve versus memory subsets of CD4^+^ helper T cells would be different at the molecular level.

To address our hypothesis, we recruited a cohort of unvaccinated participants and SARS-CoV-2 vaccinated participants between 16 December 2020 and 19 June 2021, concurrent with the vaccination rollout in England. We used Cytometry by Time of Flight (CyTOF) for phenotyping CD4 helper T cells with a panel of markers based on the Human Immunophenotyping Project (HIP) guidance [10] and identified 10 subsets with unsupervised analyses. Overall, the proportion of CD4^+^ phenotypic subsets in peripheral blood were similar between vaccinated and unvaccinated participants, an observation also found in other studies [11]. We then used fluorescence-activated cell sorting (FACS) to sort CD4^+^naïve and CD4^+^central memory (CD4^+^CM) subsets and compared the transcriptomic, epigenetic, and repertoire features with two sets of comparisons referred to as between the two subsets and within each subset by vaccination status. We combined differentially expressed genes (DEGs) from the transcriptome with the differential accessible regions (DARs) in the epigenome to ascertain genes that were poised for expression or inaccessible for transcription [12]. We observed biologically plausible differences in the transcriptomic, epigenetic, and repertoire features between CD4^+^naïve and CD4^+^CM subsets. Vaccination generated unique transcriptomic responses in both CD4^+^naïve and CD4^+^CM subsets, with biologically plausible changes to the epigenetic landscape (such as differential accessibility to the Absent in Melanoma (*AIM2*) gene) and in T cell receptor (TCR) repertoire characteristics (such as increases in clonality within the vaccinated group compared with the unvaccinated).

## 2. Materials and Methods

### 2.1. Vaccinated Donor Samples and Measurements

Plasma donors (*n* = 51) were recruited as part of CONES (Convalescent Plasma Donor Vaccine Study)-VELVET study (a study of anti-SAR-CoV-2 antibody responses in plasma donors) (Figure 1).

Blood cones were collected from NHSBT Plasma Centre, Stratford, London. Donors were divided into those vaccinated with two doses of vaccine (*n* = 26), one dose of vaccine (*n* = 17), or unvaccinated (*n* = 8).

### 2.2. Sample Collection and Processing for PBMC Isolation

Peripheral blood mononuclear cells were isolated from leukocyte reduction system chamber. Blood cones were (approximately 10 mL) emptied into a 50 mL Falcon tube. A total of 30 mL of PBS was added per 10 mL of blood. A total of 15 mL of Lymphoprep (Stemcell, Cambridge, UK, 07851) was added to a new 50 mL Falcon Conical Centrifuge tube (Thermo Fisher Scientific, Loughborough, UK, 352070). Half of the blood–PBS mixture was slowly and carefully poured on top. This was repeated for another 50 mL Falcon tube. The tubes were centrifuged at 800× *g* for 20 min at room temperature, without brakes. The PBMC layer was extracted and placed in a new Falcon tube and washed with PBS topped up to 50 mL. The tubes were centrifuged at 500× *g* for 10 min (room temperature, Acc; 9, Brake; 9). The supernatant was removed. Cells were counted and resuspended in freezing media (fetal calf serum (FCS), 10% DMSO). Cell vials were stored in Nalgene Cryogenic Freezing Container (Thermo Fisher Scientific, Loughborough, UK, 5100-001) overnight at −80 °C prior to long-term storage at −80 °C.

### 2.3. CyTOF for Phenotyping

PBMCs were suspended in 1 mL of cell staining buffer (CSB). A total of 2–3 × 10^6^ cells were washed and incubated with barcoding master mix at room temperature for 20 min. Once barcoded, cells were washed with PBS and stained with cisplatin for live/dead discrimination at a concentration of 5 μM for 60 s before being quenched in 4 mL CSB. Cells were then washed in CSB and resuspended in 45 μL of CSB and 5 μL of TruStain FcX (Biolegend, London, UK, 422302), then incubated for 10 min. For surface phenotyping, 19 antibodies were used (Table 1). A total of 50 μL of antibody master mix was added and incubated at room temperature for 30 min, vortexing every 10 min. Cells were then washed twice in CSB and fixed with 1.6% formaldehyde (Thermo Fisher Scientific, Loughborough, UK, 28908) with 10 min incubation. Fixed cells were centrifuged at 800× *g* for 5 min and washed in CSB. Before measurement, the cells were washed once in CSB and twice in cell acquisition solution (CAS) (Fluidigm, Cambridge, UK, 201244). Resuspended in CAS, counted using the countess II automatic cell counter, and adjusted to a concentration of 0.5 × 10^6^ cells/mL. Cells were spiked with EQ four-element calibration beads (Fluidigm 201078) to facilitate normalization. Files were exported as FCS files for analyses.

### 2.4. Cell Sorting of T Cell Populations

Cell vials were gently thawed at 37 °C, then washed in 40 mL of warm RPMI by centrifugation at 500× *g* for 5 min. Proteins were removed from pellet by washing with PBS. The pellet was resuspended in 100 µL PBS. An antibody master mix was made using the markers shown in Table 2, and 100 µL was added to the cell pellet. Cells were stained (in darkness) at 37 °C for 15 min, followed by room temperature for 30 min. Cells were washed in FACs buffer and passed through a strainer to remove clumps. A 1:30 dilution of live dead stain (DAPI, BD Biosciences, Wokingham, UK, 564907) was added to cells and left to incubate for 10 min. Cells were kept on ice until acquisition.

BD FACSAria (2/3) was used to sort two CD4^+^ subsets: naïve CD4^+^ T cells (CD45RA^+^, CCR7^+^), central memory T cells (CD45RA^−^, CCR7^+^), as per gating strategy reported in Appendix A. Sorted cells were stored in RLT buffer (Qiagen) at −80 °C.

### 2.5. RNA Isolation and Sequencing

RNA was isolated using Qiagen mini plus RNeasy kit (Qiagen, Manchester, UK, 47104)(manufacturer’s instructions). Libraries were prepared using the NEBNext Ultra II Directional RNA Library Prep Kit (New England Biolabs, Hitchen, UK, E7760) (manufacturer’s instructions). RNA-seq was performed on naïve and memory T cell populations using 10 ng of RNA as starting material. Libraries were sequenced as a single plate of 51 samples (27 participants) on one Illumina NovaSeq (Azenta Life Sciences, Oxford, UK) S4 lane (150 bp, paired-end sequencing) to a depth of 40 M paired-end reads.

### 2.6. DNA Isolation and ATAC-Sequencing

ATAC-seq was performed on two cell populations: naïve and memory T lymphocytes. Samples from participants who did not have any documented previous infection with COVID-19 were used for the ATAC-seq experiments. For the CD4^+^naïve cells, five samples were unvaccinated, eight samples were vaccinated with one dose, and seven samples were vaccinated with two doses. For the CD4^+^CM cells, five samples were unvaccinated, nine samples were vaccinated with one dose, and seven samples were vaccinated with two doses. All samples were CD4^+^naïve–CD4^+^CM paired except one CD4^+^naïve one dose sample that failed during sequence analysis processing. Library preparation was conducted according to the protocol by Buenrostro et al. (2015) [13], with adjustments according to the Omni protocol [14].

Briefly, 50,000 cells were washed with ice-cold PBS. The cell pellet was resuspended in ice-cold lysis buffer and incubated on ice and washed again with Wash buffer. Supernatant was discarded, and nuclei pellet was resuspended gently in transposition reaction mix and incubated at 37 °C for 30 min on a thermomixer at 1000 RPM. DNA was isolated using the Qiagen MinElute Reaction Cleanup Kit (Qiagen, Manchester, UK, 28204) and eluted in nuclease-free water. Library preparation was conducted according to the protocol by Buenrostro et al. (2015) [13], with adjustments according to the Omni protocol [14], as per manufacturer instructions. AMPure XP beads (Beckman Coulter, High Wycombe, UK, A6388) were used for library purification. Double-sided bead purification was implemented. Purified libraries contained in the supernatant were transferred to a new tube, and library quality and quantity were assessed by High Sensitivity D1000 Tapestation (Agilent, Milton Keynes, 5067-5584, 5067-5585) and Qubit fluorometer (Thermo Fisher Scientific, Loughborough, UK), respectively. Libraries were sequenced by Genewiz (Azenta Life Sciences, Oxford, UK) on a Illumina NovaSeq S4 lane platform in 2 × 150 bp configuration to a depth of 50 M paired-end reads (Additional details in Appendix A).

### 2.7. TCR Repertoire Sequencing

For TCRA and TCRB profiling, libraries were prepared using 35 ng of PBMC RNA with RIN values > 7.0 using the SMARTer Human TCR profiling kit (Takara Bio, London, UK, 634479), according to the manufacturers’ instructions using the 5′RACE technology. The initial PCR step amplified the cDNA libraries, and the second PCR step introduced the second sample barcode and Illumina TruSeq adapter sequences. AMPure XP beads (Beckman Coulter, High Wycombe, UK, A6388) were used for purification of amplified libraries. Libraries were quantified using Qubit (DNA) and Tapestation (high-sensitivity DNA). Each final library pool was spiked with 10% Phix to increase sample diversity. Paired-end sequencing was performed using the Illumina MiSeq 2 × 300 bp sequencing platform.

## 3. Analysis

### 3.1. CyTOF Analysis

Using the Helios mass cytometer (Fluidigm, Cambridge, UK), the output FCS files were normalized against EQ beads spiked into the samples [15], using CyTOF software V7.0. The normalized FCS files were passed into the R studio and debarcoded using the package *Catalyst* [16]. We used *Cytobank* [17] for manual gating analysis to clean the data using gaussian parameters, identify live intact cells, and identify CD4^+^ T cells for downstream unsupervised analysis. Major immune cell populations were identified by a biaxial gating strategy of phenotypic markers on the live, intact cells. The CD3 vs. CD19 plot identified T cells as CD3^+^ and CD19-cells. Within the T cell population, TCRgd was plotted against CD3 to identify abTCR T cells as gdTCR-subsets. abTCR T cells were further gated by plotting CD4 vs. CD8. T helper (Th) cells were identified as CD4^+^ and CD8^−^. These major populations were exported as FCS files to analyze with an unsupervised pipeline phenotype CD4^+^ subsets. For each cell subset, FCS files were exported from *Cytobank* and loaded into Rstudio. Data were transformed using hyperbolic arcsine (Arcsinh) with a co-factor of 5, and samples with less than 1000 events were removed. All cells in the dataset were clustered into cell populations using *FlowSOM* [18] within the *cytofkit2* [19] package, with the number of clusters decided by the expected number of populations based on The Human Immunology Project [10] guidance. The markers used for *FlowSOM* clustering and uniform manifold approximation and projection (UMAP) [20] analyses for the CD4^+^ subset can be found in Table 1. The Shapiro–Wilk test was used to assess normality of cell populations and median metal intensity (MMI), a surrogate for expression. An unpaired t-test (Wilcoxon) was used to determine significant differences between conditions.

### 3.2. RNA-Sequencing Analysis

Cancer Genomics Cloud (CGC) was used for initial data analysis. The raw RNA-sequencing files were uploaded to the CGC, as well as transcriptome files and transcript annotations from GENCODE (www.gencodegenes.org/human, accessed on 23 October 2023). The raw files were trimmed using *Trimmomatic* [21] to remove adapters and indexes. Quality control checks were performed on the trimmed files using *FastQC* [22]. Rabix Composer was used to generate a visual workflow for *Salmon index*, *Salmon quantification* [23], and *DESeq2* [24] analysis. *Salmon quant* files were subsequently exported, and differential expression analysis was conducted in R studio using *DESeq2*.

### 3.3. ATACseq Analysis

The Nextflow [25,26] core atacseq (v2.1.2) [27] pipeline was used for the pre-processing, genome alignment using alignment QC, enrichment analysis, peak calling, and multiQC (v1.14). The raw read QC was performed using FastQC (v0.11.9) [22], adapter trimming was achieved with Trim Galore! (v0.6.7) [28], and alignment was done using BWA [29]. Duplicates were marked and merged with the same samples using picard (v3.0.0) [30]. Filtering and removal of mitochondrial DNA, duplicates, and unmapped regions was performed using SAMtools (v1.17) [31], reads containing >4 mismatches and those that were soft-clipped were filtered out using BAMtools (v2.5.2) [32], and alignment QC was measured using picard. Genome-wide enrichment was achieved with deepTools (v3.5.1) [33], and broad peak calling was performed using MACS2 (v2.2.7.1) [34]. The peaks were annotated relative to gene features using HOMER (v4.11) [35], and the differential accessibility analysis, PCA, and clustering were performed using R (v4.3.2) [36] and DESeq2 (v1.44) [24]. For individual analysis of the samples, the BAM files were imported into SeqMonk (v1.48.1) [37] for MACS peaks calling, DESeq2 analysis, and EdgeR (v4.2.1) [38] analysis, producing annotated probe reports and BED [39] files for motif analysis using the MEMEsuite (v5.5.4) [40,41] program.

### 3.4. TCR Repertoire Analysis

MiXCR (v4.5.0) [42] was used to align VDJ regions to the reference genome and assemble CDR3 regions. Following this, Immunarch (v0.9.0) [43] was used to determine the number of clones detected and CDR3 length using the repExplore function. Diversity was calculated using the repDiversity function, and clonality was determined using the repClonality function in Immunarch [43]. All graphs were visualized using ggplot2 (v3.5.0) [44]. We used MiXCR to deconvolute T cell alpha and beta receptor identities and analyzed using TCR repertoires using Immunarch to establish clonotype number, diversity, CDR3 length, clonal abundance, and the top 25 clones in naïve and memory CD4^+^ T cells based on vaccination status.

## 4. Results

### 4.1. Study Cohort

Our cohort consisted of unvaccinated (*n* = 8), one dose vaccinated (*n* = 17), and two dose vaccinated (*n* = 26) participants. Individuals were checked for evidence of SARS-CoV-2 infection by serology test. Demographic features of participants and the vaccinations received reflected the vaccination schedule in England, United Kingdom, at the time of study recruitment (Table 3).

### 4.2. Comparisons of Phenotypic Changes in CD4^+^ T Cell Subsets between Groups

We identified 10 CD4^+^ subsets mappable to the CD4^+^ compartment highlighted in the HIP guidance (Figure 2a,b). CD4^+^Naïve and CD4^+^ Th1 were the most common subsets in unvaccinated, single-dose vaccinated, and two-dose vaccinated groups (median proportions were 43.5%, 45%, and 41.8% for naïve and 17.9%, 15.8%, and 17.2% of Th1 subsets, respectively). There were two CD4^+^ Treg subsets (CD4^+^ Treg naïve and Treg memory), and the CD4^+^ Treg memory subset in the two-dose vaccination group was higher compared with the single-dose vaccination group (Figure 2b,c). There were three CD4^+^CM and CD4+effector memory (CD4^+^EM) subsets (Figure 2b,c): CD4^+^CM1 (CXCR5^+^), CD4^+^CM2 (CD38^+^), CD4^+^CM3 (CD127^−^), and CD4^+^EM1 (CD25^+^CCR4^+^CD127^+^), CD4^+^EM2 (CD25^−^CCR4^−^CD28^−^), and activated CD4^+^EM3 (CD38^+^HLA-DR^+^, respectively). The CD4^+^CM2 (CD38^+^) subset was significantly higher in the one-dose vaccinated group compared with the unvaccinated group. Further, MMI of CD4^+^ T cell activation markers (CD38/HLA-DR) were higher in the CD4^+^CM subsets compared with the CD4^+^naïve subset, and the HLA-DR MMI was higher in the vaccinated group compared with the unvaccinated group (Figure 2d).

### 4.3. Comparisons of Molecular Features in Naïve and Central Memory T Helper Cell Subsets

We sorted CD4^+^naïve and CD4^+^CM subsets using FACS (Appendix A) to measure transcriptome, epigenome, and TCR repertoire. To address our hypothesis, we performed two sets of comparisons: (a) *within* group comparisons and (b) *between* group comparisons (Table 4).

### 4.4. Within-Group Comparisons

Within-group comparisons were performed to highlight the molecular differences between the CD4^+^naïve compared with the corresponding CD4^+^CM subset.

#### 4.4.1. Transcriptome

Overall, we observed 205 DEGs, with 181 unique to vaccinated, 20 unique to unvaccinated, and four overlap DEGs between the groups (Figure 3b). In the overall vaccinated group comparison, the CD4^+^naïve subset had 185 DEGs (85 upregulated and 100 downregulated genes), compared with the CD4^+^CM subset. In the unvaccinated group, the CD4^+^naïve subset had 24 DEGs (eight upregulated and 16 downregulated genes), compared with the CD4^+^CM subset (Figure 3a,b). In the single-dose vaccinated group, the CD4^+^naïve subset had 60 DEGs (40 upregulated and 20 downregulated genes), compared with the CD4^+^CM subset. In the two-dose vaccinated group, the CD4^+^naïve subset had 53 DEGs (24 upregulated and 29 downregulated genes), compared with the CD4^+^CM subset. In the unvaccinated group, no biological pathways were enriched. In the vaccinated group, the top 10 pathways enriched included biologically plausible mechanisms such as cytokine signaling, Tregs development, and viral protein interaction with cytokines (Figure 3c).

#### 4.4.2. Epigenome

In the overall vaccinated group, we observed 166 probes (123 genes, including the genes observed in the two dose vaccinated groups) in DARs, amongst the 29,872 MACS peaks between the CD4^+^naïve and CD4^+^CM subsets (Figure 3d). In the unvaccinated group, there were no differences in the DARs between CD4^+^naïve and CD4^+^CM subsets (Appendix A). In the single dose vaccinated group, we observed the *AIM2* gene in the DARs, amongst the 10,745 MACS peaks between the CD4^+^naïve and CD4^+^CM subsets. In the two-dose vaccinated group, we observed 11 probes that make up the following eight genes: *AIM2*, niban apoptosis regulator-1 (*NIBAN*), cryptochrome circadian regulator 1 (*CRY1*), shieldin complex subunit 1 (*SHLD1*), muskelin 1 (*MKLN1*), Adaptor Related Protein Complex 4 Subunit Sigma 1 (*AP4S1*), Long Intergenic Non-Protein Coding RNA 3044 (*LINCO044*) in DARs, amongst the 14,924 MACS peaks between the CD4^+^naïve and CD4^+^CM subsets.

In the motif analysis of the overall vaccinated group comparing CD4^+^naïve vs. CD4^+^CM subsets, we identified three motifs, predicting three transcription factors (TF) for each motif, namely (*ZN281*, *ZN770*, *ZN223*) for the first motif, (*ZN362*, *Z585A*, *ZN613*) for the second, and (*ZN560*, *ZBT49*, *ZN441*) for the third motif (Figure 3e) linked to the genes observed in the DARs (Figure 3d). Gene ontology analysis of the overall CD4^+^naïve vs. CD4^+^CM subset comparison identified pathways linked to the cytoplasm, cell development, and the purinergic receptor (*P2 × 7*) signaling complex (Figure 3f). We could not perform motif analysis in the unvaccinated group comparison of CD4^+^naïve vs. CD4^+^CM subsets due to no DARs and in the single dose group comparison of CD4^+^naïve vs. CD4^+^CM subsets due to only one DAR. In the motif analysis of the two-dose vaccinated group comparison of CD4^+^naïve vs. CD4^+^CM subsets, we identified two motifs, predicting three transcription factors (TF) for each motif, namely homeobox (HB) gene family clusters named A, B, C, and D (*HXA13*, *HXC13*, *HXD13*) for one motif and zinc finger protein *(ZNF) (ZNF586*, *Z585B*, *ZN675)* for the other (Figure 3e) linked to the genes observed in the DARs (Figure 3d). *HXA1*, *HXC13*, and *HXD13* are a family of homeobox genes and are all linked to early development [45]. The zinc finger proteins *ZN586*, *Z585B*, and ZN675 are linked to regulation of transcription by RNA polymerase II [46]. Gene ontology analysis of the CD4^+^naïve vs. CD4^+^CM subset comparison identified one enriched transcription factor, *ZNF75D*, in the two dosed vaccinated groups (Figure 3f).

The overall vaccinated genes in DEGs overlayed with the overall vaccinated DEGs gave eight genes (Figure 3g,h); *NIBAN1*, *AIM2*, *CD58*, Chromodomain Helicase DNA Binding Protein 2 (*CHD2)*, Coagulation Factor II Thrombin Receptor (*F2R*), *ZNF365*, ITGB1 Adjacent Tumor Promoting LncRNA (*IATPR*), and a Disintegrin and Metalloproteinase Domain 19 (*ADAM19*). These genes play roles in apoptosis regulation [47], inflammasome regulation [48], antiviral response regulation, associated with autoimmune diseases and inflammation [49], translation regulation, and chromatin remodeling [50]. They are also involved in tissue injury repair [51], antibacterial function [52], tumor promotion [53], and *ADAM19* is linked to the regulation of human dendritic cells [54], respectively. All located in an open region and with reduced expression, apart from *CHD2*, which had an increase in expression (Figure 3h). When comparing one-dose vaccinated DARs on top of DEGs, they gave zero overlapped genes (Figure 3h); however, the two-dose vaccinated DARs overlayed on the DEGs gave three genes, all located in the open region and with reduced expression: *ZNF365*, *IATPR*, and *ADAM19*.

### 4.5. Between Group Comparisons

We performed two between-group comparisons to highlight the molecular differences due to vaccination (i) within the CD4^+^naïve subset and (ii) within the CD4^+^CM subsets.

#### 4.5.1. Between Group Comparisons of Molecular Mechanisms in CD4^+^Naïve Subset

##### Transcriptome

The CD4^+^naïve subset in the overall vaccinated group had 24 DEGs (all upregulated), compared with the unvaccinated group. The CD4^+^naïve subset in the one-dose vaccinated group had 27 DEGs (24 upregulated and three downregulated) **(**Figure 4a), compared with the unvaccinated group. In contrast to the one-dose vs. unvaccinated comparison, the CD4^+^naïve subset in the two-dose vaccinated group had only three DEGs (all upregulated), compared with the unvaccinated group. Importantly, there were no DEG in the CD4^+^naïve subset with the one dose vs. two dose comparisons. Enriched pathways in the CD4^+^naïve subset in vaccinated participants were lymphocyte response to vaccination, alongside pathways associated with glial cells and synapses (Figure 4d).

##### Epigenome

In the CD4^+^naïve subset, across all comparisons, we consistently observed two statistically significant probes of the same gene (both upregulated (*RNA5SP162*), RNA 5S Ribosomal Pseudogene 162) (Figure 4e) in the DARs amongst 66,471, 24,643, and 34,631 MACS peaks when comparing the unvaccinated with overall vaccinated, one dose and two dose group comparisons, respectively. The 5S rRNA in general is associated with enhancing protein synthesis through stabilization of the ribosome structure [55].

#### 4.5.2. Between Group Comparisons of Molecular Mechanisms in CD4^+^CM Subset

##### Transcriptome

The CD4^+^CM subset in the overall vaccinated group had 20 DEGs (17 upregulated and three downregulated), compared with the unvaccinated group. In the CD4^+^CM subset, the one-dose vaccinated group had 44 DEGs (34 upregulated and 10 downregulated), compared with the unvaccinated group (Figure 4b). In contrast to the one-dose vs. unvaccinated comparison, the CD4^+^CM subset in the two-dose vaccinated group had only one downregulated DEG compared with the unvaccinated group. The same downregulated DEG was also observed in the CD4^+^CM subset with the one-dose vs. two-dose comparison. Enriched pathways in the CD4^+^CM subset in vaccinated participants were associated with cytokine signaling, apoptosis, and cell structure/adhesion, to highlight a few (Figure 4d).

##### Epigenome

In the CD4^+^CM subset, there were no differences in the DARs between the unvaccinated and overall vaccinated group comparisons. However, we observed two statistically significant upregulated probes, Glutamine Amidotransferase Class 1 Domain Containing 3 (*GATD3*), the regulator of innate and adaptive immunity, driving differentiation of Th2 cells [56] in the DARs amongst 13,826 MACS peaks in the one dose-vaccinated vs. unvaccinated comparison (Figure 4f). Finally, in the CD4^+^CM subset, there were no differences in the DARs between the unvaccinated and two-dose vaccinated group comparisons.

In summary, with the between-group comparisons, we infer that the molecular responses are unique to the subsets, based on limited overlap of DEGs (two DEGs overlapping between the CD4^+^naïve and the CD4^+^CM subsets) and no overlap in the DARs in response to vaccination (Figure 4c).

### 4.6. Comparisons of Repertoire Features between Unvaccinated, Single-Dose Vaccinated, Two-Dose Vaccinated, and Overall Vaccinated Groups

#### 4.6.1. TCRA Comparisons

CD4^+^CM subsets in all groups did not have a notable difference in the number of clones (Figure 5a), compared with the corresponding CD4^+^naïve subsets. However, the CD4^+^CM subsets in all four groups had lower diversity compared with their corresponding CD4^+^naïve subsets (Figure 5b).

In the CD4^+^naïve subset, amongst the unvaccinated, single-dose, and overall vaccinated, 90% of the clones were rare clones (Figure 4e). In contrast, the CD4^+^naïve subset, amongst the two-dose vaccinated, only 50% of the clones were rare clones. In contrast, in the CD4^+^CM subset, clonal distribution was similar across all four groups.

In the unvaccinated group, amongst the top 25 clones identified in the CD4^+^CM subset, six clones were also present in the CD4^+^naïve subset (Figure 4g). In the single-dose vaccinated group, amongst the top 25 clones identified in the CD4^+^CM subset, 16 clones were also present in the CD4^+^naïve subset. In the two-dose vaccinated group, amongst the top 25 clones identified in the CD4^+^CM subset, 20 clones were also present in the CD4^+^naïve subset.

#### 4.6.2. TCRB Comparisons

CD4^+^CM subsets in the unvaccinated and in the single dose vaccinated had lower numbers of clones compared with the corresponding CD4^+^naïve subsets (Figure 5c). This difference was not significantly observed in the two dose or overall vaccinated comparisons between CD4^+^naïve and CD4^+^CM subsets. The memory subsets in all four groups had lower diversity compared with their corresponding CD4^+^naïve subsets, and we observed significantly lower diversity in the unvaccinated, one-dose, and overall vaccinated CD4^+^CM compared with the unvaccinated, one-dose, and overall vaccinated CD4^+^naïve subsets, respectively (Figure 5d).

In the CD4^+^naïve subset, clonal distribution was between ~85–90% of the clones were rare clones in all four groups (Figure 5f). In contrast, in the CD4^+^CM subset, clonal distribution was between ~65–70% across all four groups.

In the unvaccinated group, amongst the top 25 clones identified in the CD4^+^CM subset, 10 clones were also present in the CD4^+^naïve subset (Figure 5h). In the single-dose vaccinated group, amongst the top 25 clones identified in the CD4^+^CM subset, nine clones were also present in the CD4^+^naïve subset. In the two-dose vaccinated group, amongst the top 25 clones identified in the CD4^+^CM subset, nine clones were also present in the CD4^+^naïve subset.

#### 4.6.3. TCR-Matched Epitopes

CD4^+^CM subsets in the unvaccinated group matched 21 unique epitopes linked to 12 antigens, of which five antigens were found in SARS-CoV-2 (Appendix A). CD4^+^CM subsets in the one-dose vaccinated group matched 25 unique epitopes linked to 13 antigens, of which seven antigens were found in SARS-CoV-2. CD4^+^CM subsets in the two-dose vaccinated group matched 13 unique epitopes linked to nine antigens, of which six antigens were found in SARS-CoV-2.

## 5. Discussion

In COVID-19 naïve participants, the key phenotypic responses of vaccination on CD4^+^ T cells included CD4^+^ T cell activation, expansion of circulating CD4^+^ memory subsets, and increase in Treg memory subsets. This response has similarities with a natural infection [57]. Expansion of Treg memory is biologically plausible, given its role in immunological homeostasis, particularly regulating effector cytokine responses [58], which could lower the risk of severe disease in SARS-CoV-2 infections [59]. There were several key differences in the transcriptomic profiles between CD4^+^naïve and CD4^+^CM cell subsets and enrichment of pathways such as “cytokine signaling” and “viral protein interaction with cytokine” that are associated with effector cytokine response following immune cell activation [60,61]. Comparing the DAR between CD4^+^naïve and CD4^+^CM subsets by vaccine status, we observed 166 differentially accessible regions in the vaccinated group. The most significant of these regions, increased in the CD4^+^CM subsets, were linked to the zinc finger protein TFs that regulate the transcription by RNA polymerase II, an enzyme the SARS-CoV-2 virus and mRNA vaccines rely on [62]. Gene ontology analysis of the combined vaccinated group enriched the *P2* × *7* signaling complex pathway. This receptor triggers a cytokine release [63] and is linked to the expression and release of pro-inflammatory cytokines and has been found to be elevated in the serum of COVID-19 patients [64]. The DAR from the overall vaccinated group identified eight genes, seven of which were downregulated in expression and one gene—*CHD2*—was upregulated. These downregulated genes were linked to translation regulation, apoptosis regulation, inflammation regulation, and inflammasome activation, and the upregulated gene was linked to chromatin remodeling. The open regions at these genes could suggest the COVID-19 vaccine reduces inflammation and the release of pro-inflammatory cytokines; at the same time, immune cells are poised for expression.

In the transcriptomic differences between the vaccinated and the unvaccinated groups comparisons, most changes occurred between the unvaccinated and one-dose vaccination groups in CD4^+^naïve and CD4^+^CM subsets. The DEG were mostly unique to each subset, with features such as lymphocyte responses to vaccination only in the CD4^+^naïve subsets and inflammation/cytokine signaling only in the CD4^+^CM cell subsets.

Amongst the minimal epigenetic changes between unvaccinated and vaccinated in the CD4^+^naïve subset and the CD4^+^CM subset, the DAR identified were linked to protein synthesis and regulation of Th2 differentiation, respectively. The Th2 cytokines have been linked to a reduced risk of severe disease in SARS-CoV-2 infection via inhibition of viral entry into the cells by increasing epithelial mucin secretion [65]. This function is a desired response in the CD4^+^CM vaccinated group in aiding longer-term protection against the virus.

As the majority of mature T cells express TCRA or TCRB isoforms and form multiprotein complexes with CD3 chains [66] (Appendix A), we explored the changes within the TCRA and TCRB isoforms between the CD4^+^naïve and CD4^+^CM cells by vaccine status. We specifically looked for SARS-CoV-2 clonal expansion [67]. We found a lower diversity in the unvaccinated group between CD4^+^naïve and CD4^+^CM cells in both TCRA and TCRB chains, which supports the notion that vaccination generates the CD4^+^CM cells against the SARS-CoV-2 virus. In the unvaccinated, one-dose, and overall vaccinated groups, the CD4^+^naïve subset on the TCRA chain had ~90% rare clones, which dropped down to 50% of the clones in the two vaccinated groups. In the TCRA isoform, the top 25 clones found in the CD4^+^CM subset compared with the CD4^+^naïve subset increased from unvaccinated to one dose and then to two doses, suggesting less differences between the two subsets. Whereas the TCRB isoform found a greatly reduced number of clones with the unvaccinated and less with the one dose and two doses in the CD4^+^CM subset compared with the CD4^+^naïve subset, suggesting a greater difference between the two subsets and more differentiation.

Multiomic analysis of the COVID-19 vaccines on the CD4^+^ T cells provides valuable information on the DEGs occurring within the immune cells and allows us to predict which genes are poised for expression or are inaccessible for expression. This gives an advantageous insight into their functional potential [12] that can be used in future vaccine development. We have observed that although expression levels in cytokine signaling increase with vaccination, there are inaccessible regions that would ordinarily influence the regulation of inflammation and transcription; this would reduce disease severity, which supports current literature concerning the vaccine function [68]. This study supports current literature in recognizing the biologically plausible, unique function of the two immune subsets influenced by the vaccine. The first dose shows the greatest response in the transcriptome and epigenome within each subset. After the second dose, the CD4^+^CM subset shows a greater response in the epigenome compared with the CD4^+^naïve subset. It is biologically plausible that some of the molecular changes we observe could overlap between vaccines. Otherwise stated, what we observed may not be unique to SARS-CoV-2 vaccines. This is an important future research question.

Our work has key limitations. We show biologically plausible changes in DEG, DAR, and clonal expansion of SARS-CoV-2 S protein-related epitopes. We note that our cohort consisted of vaccinated individuals with a mix of mRNA and adenovirus vector vaccines, which meant that we are not reporting responses specific to a vaccine. We also note that the sample collection took place at the height of the pandemic when the social distancing restrictions were in place. Thus, the participants involved in this study were recruited primarily from a pool of plasma and blood donors, and the elderly are underrepresented in our cohort. The two-dose patients had a median age of 55.5 years, which on average was 15 years older than the single-dose group and the unvaccinated group. Increasing age increases the likelihood of acquiring new epigenetic changes. Thus, it is plausible that the difference in age could contribute to the epigenetic differences with two doses, as opposed to it being a vaccine effect. However, none of the differentially accessible regions were uniquely related to aging. Our cohort was also limited to COVID-19-naïve individuals only. Comparing the immune response to vaccination with the immune response to infection could provide insight into the differences in epigenetic changes between these conditions that generate immunological memory.

Another point to highlight is that during this time the choice and availability of vaccines were limited. It was not yet fully established the efficacy and longevity of each of the vaccines, and so the idea was not to compare the vaccines but instead to establish a better understanding of the multiomic response from the COVID-19 vaccines in general. Admittedly, single-cell multiomics would have given us more information on the variation and diversity of the immune response to the vaccination.

## 6. Conclusions

Vaccination is associated with activation of T helper cell subsets. We observed limited incremental changes in DEGs and DARs between one and two dose vaccinations, with the changes highlighting biologically plausible pathways in T cell biology modulating host response. Repertoire analyses highlight clonal expansion to SARS-CoV-2 specific epitopes.

## Figures and Tables

**Figure 1 vaccines-12-01040-f001:**
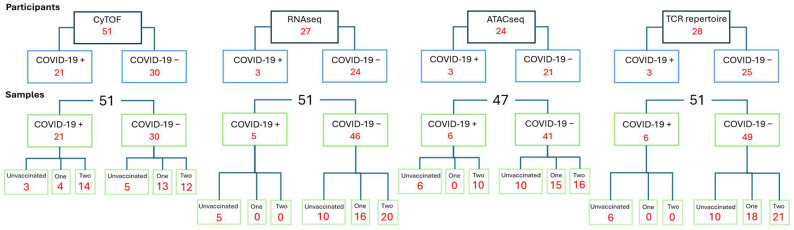
Flowchart of study. Number of participants in each experiment split by COVID-19 positive, below; number of samples within each experiment split by COVID-19 and by vaccination status.

**Figure 2 vaccines-12-01040-f002:**
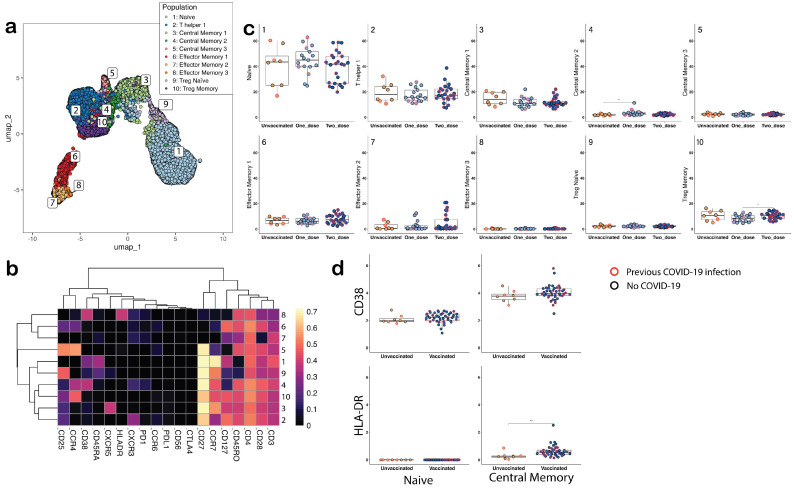
Identification of immune cell subsets assessed by time of flight mass cytometry (cyTOF) and flow cytometry gating used for sorting naïve and memory subsets. The populations were grouped into unvaccinated, one-dose-vaccinated, and two-dose-vaccinated cohorts to assess changes with vaccination status. (**a**) UMAP of 10 CD4 T cell populations from individuals that were unvaccinated and vaccinated with the COVID-19 vaccine, clustered based on marker expression: 1: Naïve (CCR7^+^CD45RA^+^CD45RO^−^CD38^+^CD25^−^CCR4^−^), 2: T helper 1 (CCR7^low^CD45RA^−^CD45RO^+^CXCR3^+^CCR6^−^), 3: Central memory 1 (CCR7^+^CD45RA^−^CD45RO^+^CCR4^−^CD127^+^CXCR5^+^), 4: Central memory 2 (CCR7^+^CD45RA^−^CD45RO^+^CCR4^+^CD25^low^CD38^+^PD1^low^), 5: Central memory 3 (CCR7^low^CD45RA^−^CD45RO^+^CCR4^+^CD25^+^CD127^−^), 6: Effector memory 1 (CCR7^−^CD45RA^−^CD45RO^+^CCR4^low^CD25^low^CD127^+^), 7: Effector memory 2 (CCR7^−^CD45RA^−^CD45RO^low^CD28^−^), 8: Activated Effector memory 3 (CCR7^−^CD45RA^−^CD45RO^+^CD38^+^HLA-DR^+^), 9: Treg naive (CCR7^+^CD45RA^low^CD45RO^low^CD25^+^CD127^low^), 10: Treg memory (CCR7^+^CD45RA^−^CD45RO^+^CCR4^+^CD25^+^CD127^+^). Bold markers in the central memory and effector memory groups highlight the differences between each group. (**b**) Heatmap of scaled marker intensity used to define the subpopulations. (**c**) CD4^+^ T cell population grouped by vaccination status. (**d**) Differences in the mean metal intensity (MMI) of proteins involved in activation (HLA-DR and CD38) between unvaccinated and vaccinated individuals in the naïve and central memory CD4^+^ subpopulations. Individuals that reported a previous COVID-19 infection are highlighted with a red ring in (**c**,**d**). Statistical analysis was performed using a *t* test. *: *p* ≤ 0.05; **: *p* ≤ 0.01.

**Figure 3 vaccines-12-01040-f003:**
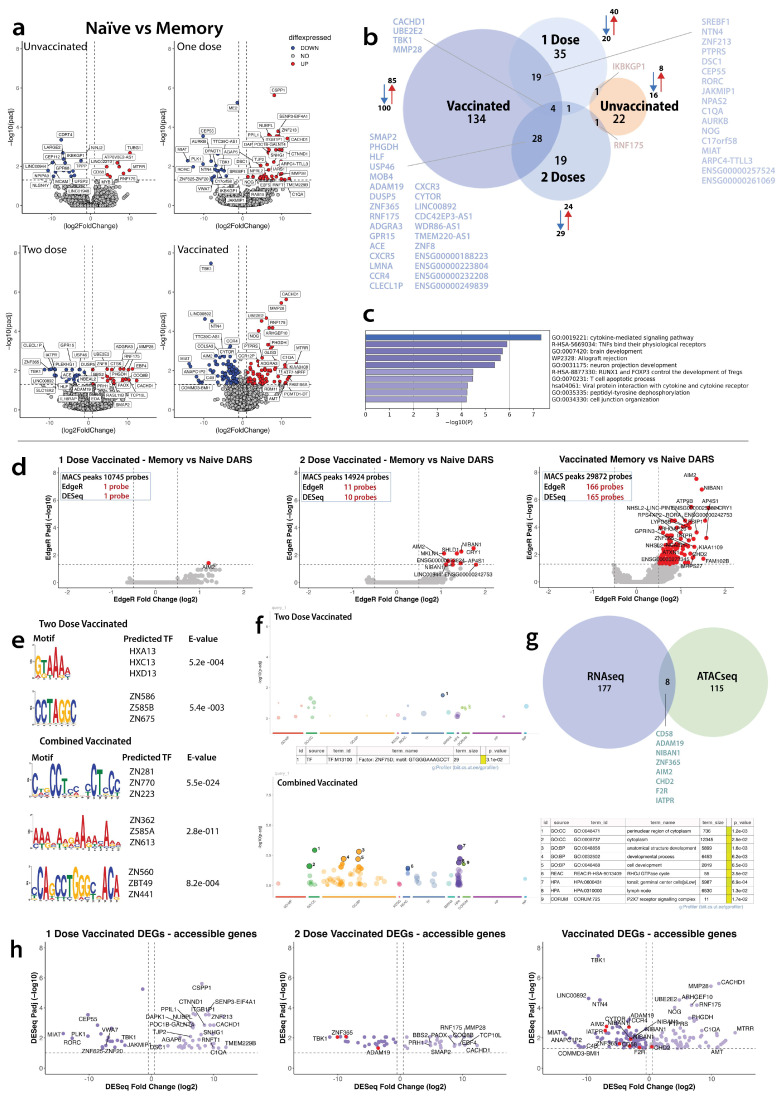
Transcriptomic and epigenetic differences between naïve CD4^+^ and central memory CD4^+^ subsets by vaccine status. (**a**) Volcano plots of differentially expressed genes (DEGs) between memory and naïve CD4 subsets. Donors were separated based on vaccination status and dosage. Upregulated genes (red) are defined as log2 fold change > 1 and adjusted *p*-value ≤ 0.05, and downregulated (blue) are defined as log2 fold change < −1 and adjusted *p*-value ≤ 0.05. (**b**) Venn diagram depicting the overlap of DEGs comparing CD4 naïve with CD4 memory in unvaccinated (orange) and vaccinated (blue) donors. Upregulated genes (red arrow) and down regulated genes (blue arrow) next to the corresponding group. (**c**) Top 10 enrichment GO terms (identified using Metascape) in the overall vaccinated group. (**d**) Volcano plots of the differentially accessible regions (DARs) of central memory CD4^+^ T cells compared with naïve. Each dot depicts a probe present within a region of a gene that has changed its accessibility using the statistical analysis; EdgeR and DESeq2. The red dot shows an increased accessible region (log2 fold change > 1 and adjusted *p*-value ≤ 0.05), there were no dots showing a decreased accessible region (log2 fold change < 1 and adjusted *p*-value ≤ 0.05). Central memory CD4^+^ T cells versus naïve T cells in (left to right) one-dose vaccinated individuals, two-dose vaccinated individuals, and combined-dose vaccinated individuals. (**e**) Motif analysis on DARs identified using EdgeR in two-dose vaccinated individuals and combined-vaccinated individuals when observing changes in naïve vs. memory T cells. Statistically significant predicted transcription factors have been listed for both groups in question. (**f**) Enriched gene ontology analysis (identified using gene profileR) of the DARs in the two-dose vaccinated and overall-vaccinated groups. (**g**) Venn diagram of overall vaccinated DEGs overlapping with the overall vaccinated genes identified in DARs. (**h**) Volcano plots of differentially expressed genes (DEGs) overlayed with DARs (left to right) in individuals that are vaccinated with one dose, vaccinated with two doses, and a combined both one dose and two doses. Each dot is a unique point along the transcriptome, filtered by an adjusted *p*-value ≤ 0.05 and labeled by overlapping genes The darker purple dots (left of the volcano plot) show reduced accessibility, the lighter purple dots show an increase accessibility. The red dots show the DEGs identified in (**d**).

**Figure 4 vaccines-12-01040-f004:**
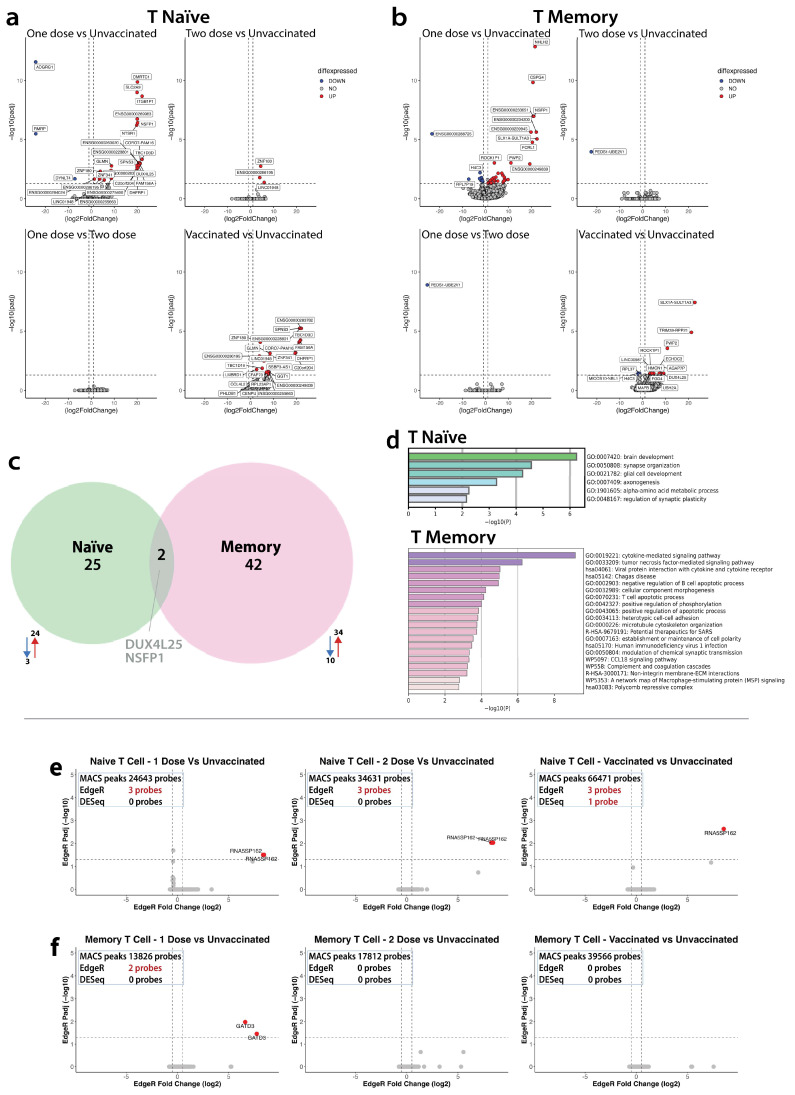
Transcriptomic and epigenetic differences within naïve CD4^+^ and central memory CD4^+^ subsets by vaccine status. (**a**) Volcano plots of DEGs in naïve CD4^+^ subsets comparing vaccination dosages. Upregulated genes (red) are defined as log2 fold change > 1 and adjusted *p*-value ≤ 0.05, and downregulated genes (blue) are defined as log2 fold change < −1 and adjusted *p*-value ≤ 0.05. (**b**) Volcano plots of DEGs in memory CD4^+^ subsets comparing vaccination dosages. Upregulated genes (red) are defined as log2 fold change > 1 and adjusted *p*-value ≤ 0.05, and downregulated genes (blue) are defined as log2 fold change < −1 and adjusted *p*-value ≤ 0.05. (**c**) Venn diagram depicting the overlap of DEGs in response to 1 dose of vaccination compared with unvaccinated donors in CD4 naïve (green) and CD4 memory (pink). (**d**) Enriched GO terms (identified using Metascape) in naïve CD4^+^ (top, green) and memory CD4^+^ (bottom, pink) vaccinated donors. (**e**,**f**) Volcano plots of the differentially accessible regions (DARs) of central memory CD4^+^ T cells compared with naïve. Each dot depicts a probe present within a region of a gene that has changed its accessibility using the statistical analysis EdgeR and DESeq2. The red dot shows an increased accessible region (log2 fold change > 1 and adjusted *p*-value ≤ 0.05), and the blue dot shows a decreased accessible region (log2 fold change < 1 and adjusted *p*-value ≤ 0.05). (**e**) Naïve T cells only, (left to right) observing differences in unvaccinated vs. one-dose vaccinated individuals, unvaccinated vs. two-dose individuals, and unvaccinated vs. combined-dose vaccinated individuals. See Appendix A. (**f**) Central memory T cells only, observing the differences in unvaccinated vs. one-dose vaccinated individuals, unvaccinated vs. two-dose individuals, and unvaccinated vs. combined-dose vaccinated individuals.

**Figure 5 vaccines-12-01040-f005:**
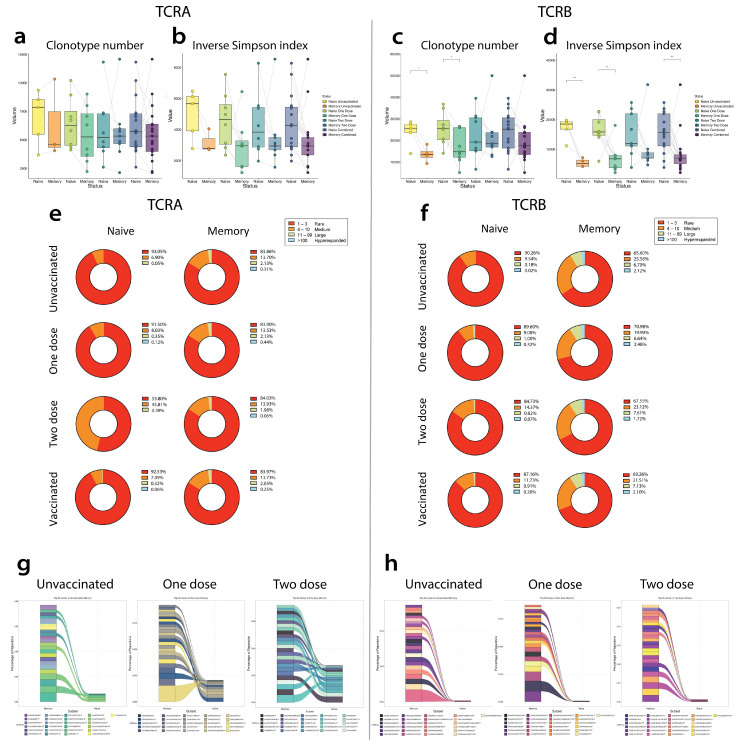
T cell repertoire changes in naïve and memory subsets. (**a**) Number of TCRα and (**c**) TCRβ clonotypes detected in naïve and memory T cells from unvaccinated, one-dose, two-dose, and overall vaccinated patients. (**b**) Diversity of TCRα and (**d**) TCRβ in naïve and memory T cells. (**e**) Average clone size was determined in naïve and memory subsets in unvaccinated, one dose, two doses, and overall vaccine in TCRα and (**f**) TCRβ chains. Clonal size was grouped by rare (1–3), medium (4–10), large (11–99), and hyperexpanded (>100). (**g**,**h**) Alluvian plot of the top 25 clones identified in the memory subset present in the naïve subset in the unvaccinated, one-dose, and two-dose vaccinated groups. See Appendix A.

**Table 1 vaccines-12-01040-t001:** CyTOF markers used for FlowSOM clustering and UMAP analyses for the CD4^+^ subset.

CyTOF Markers		
CCR6	CD25	CCR7
CD38	CD127	PD1
CXCR5	HLADR	CTLA4
CD45RA	CCR4	CD3
CD45RO	CD28	CD4
CD27	CD56	
CXCR3	PDL1	

**Table 2 vaccines-12-01040-t002:** Full panel used for fluorescence-activated cell sorting of CD4^+^ T cells.

Marker	Fluorochrome	Manufacturer	Clone	Cat Number	Volume (µL)
CD3	BUV395	BD	SK7	564,001	2
CD4	BV786	BD	SK3	563,877	2
CD45RA	BV510	BD	HI100	563,031	2
CCR7	PE-CF594	BD	150503	562,381	4
CD19	FITC	BD		555,412	2
CD27	BV711	Biolegend	O323	302,834	2
IgG	APC	BD		550,931	5
IgM	PE	BD		555,783	10
Live Dead	DAPI (1 mg)	BD		564,907	2 µL of a 1:30 dilution

**Table 3 vaccines-12-01040-t003:** Participant characteristics and clinical data.

Characteristic	Unvaccinated	One Dose	Two Dose
	(*n* = 8)	(*n* = 17)	(*n* = 26)
Age, median (IQR)	37 (23–53)	35.5 (32.25–39.75)	55.5 (48–60)
Male *n* (%)	8 (100%)	17 (100%)	25 (96.2%)
**Previous SARS-CoV-2 Infection, *n* (%)**			
Yes	3 (37.5%)	4 (23.5%)	13 (50%)
No	5 (62.5%)	14 (82.3%)	13 (50%)
**Symptoms, *n* (%)**			
Loss of smell	2 (66.6%)	1 (25%)	6 (46.2%)
Cough	2 (66.6%)	3 (75%)	7 (53.8%)
Loss of appetite	2 (66.6%)	1 (25%)	5 (38.5%)
Fatigue	3 (100%)	4 (100%)	11 (84.6%)
Chest pain	0	0	2 (15.4%)
Sore throat	2 (66.6%)	1 (25%)	1 (7.7%)
Muscle pain	3 (100%)	3 (75%)	8 (61.5%)
Hoarseness	2 (66.6%)	0	0
Abdominal pain	0	0	0
Headache	2 (66.6%)	3 (75%)	8 (61.5%)
Fever	1 (33.3%)	0	4 (30.8)
Shortness of breath	1 (33.3%)	0	5 (38.5%)
Diarrhoea	0	0	1 (7.7%)
Confusion	0	0	1 (7.7%)
Other	0	1 (25%)	1 (7.7%)
Brand of vaccine first dose, *n* (%)	n/a	17	26
Pfizer		11 (64.7%)	8 (30.8%)
Astra Zeneca		5 (29.4%)	18 (69.2%)
Moderna		2 (11.7%)	0
Brand of vaccine second dose, *n* (%)	n/a	n/a	26
Pfizer			8 (30.8%)
Astra Zeneca			17 (65.4%)
Moderna			0
NA			1 (3.8%)
Time between vaccination and sampling, median (IQR) days	n/a	31 (20.5–42.75)	47.5 (28.25–78.5)

**Table 4 vaccines-12-01040-t004:** Comparisons performed.

Within Group Comparisons of CD4^+^Naïve vs. CD4^+^CM	
Comparison within unvaccinated group	Unvaccinated: CD4^+^naïve vs. CD4^+^CM
Comparisons within vaccinated group	One dose vaccinated: CD4^+^naïve vs. CD4^+^CM Two dose vaccinated: CD4^+^naïve vs. CD4^+^CM Overall vaccinated: CD4^+^naïve vs. CD4^+^CM
**Between Group Comparisons Vaccinated vs. Unvaccinated by the CD4^+^Naïve and CD4^+^CM Subset**	
CD4^+^Naïve subset	Vaccinated CD4^+^naïve (one dose) vs. Unvaccinated CD4^+^naïve Vaccinated CD4^+^naïve (two dose) vs. Unvaccinated CD4^+^naïve Vaccinated CD4^+^naïve (overall) vs. Unvaccinated CD4^+^naïve
CD4^+^CM subset	Vaccinated CD4^+^CM (one dose) vs. Unvaccinated CD4^+^CM Vaccinated CD4^+^CM (two dose) vs. Unvaccinated CD4^+^CM Vaccinated CD4^+^CM (overall) vs. Unvaccinated CD4^+^CM

## Data Availability

The original contributions presented in the study are included in the article/Appendix A. Further inquiries can be directed to the corresponding author(s).

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
