# Peer review of "Changes in Phenotypic and Molecular Features of Naïve and Central Memory T Helper Cell Subsets following SARS-CoV-2 Vaccination"

_vaccines, 2024, doi:10.3390/vaccines12091040_

Round 1

Reviewer 1 Report

Comments and Suggestions for Authors
Major revision:
1. Add a flow-chart of the study
2. The background would benefit of 1-2 comments about the role of epigenetics in COVID-19 in order to strenght the hypothesis (quote: PMID: 37781451) as well as additional recent works about the phenotyping of CD4+ T cells after one/two dose vaccination (quote: PMID: 35114198, PMID: 34726470, PMID: 35003131).

3. Did the Authors perform a power analysis to justify the sample size?

4. The Authors declare n=51 plasma donors but the sum of three gropus (2 doses, 1 dose and no dose of vaccination) is 52. Check.

5. Line 136-139. Please, clarify more in detail which is the meaning of "a single pool of 51 samples (27 participants)". Which patients? Which cells? etc....

6. Please, clarify which samples/patients were used for ATAC-seq experiments.

7.Figures are unreadable in many panels. 

8. Lines 526-528. Please, rephrase for clarity. 

Comments on the Quality of English Language

To be improved. 

Reviewer 2 Report

Comments and Suggestions for Authors

The paper reports some interesting results about COVID vaccination especially with respect to the HLA-DR activation and the modulation of inflammatory gene expression by DEG and DAR, as well as TCR clonality. It also gives a master class in the analysis of T cell subsets with an excellent presentation of the methodology.

The description of the subjects in Table 3 appears to need correction or explanation with respect to the percentage of subjects with symptoms in the unvaccinated group. The loss of smell for 2/3 subjects is 60 not 40% and so on (the vaccinated groups are OK).

Two important considerations that have not been dealt with are the time between the vaccination and sampling and any effect of the SARSCoV-2 infections.

Were the subjects checked for evidence of SARSCoV-2 infection by serology?

A discussion point is whether or not the changes described are specific for COVID vaccination or vaccination in general.

Reviewer 3 Report

Comments and Suggestions for Authors

Mosavie et al. present a sound technical exercise with very limited biological value.

Some of the limitations like the extreme heterogeneity of the very small groups exposed or naïve to COVID-19 treated with Pfizer, Astra Zenica, or Moderna vaccine was mentioned. 

There are a number of further limitations not mentioned

Unfortunately, the most interesting group – elderly over 70 years of age – was not considered. This population profited most from the (mRNA) vaccine. What was the reason?

The CD4 T cells are important cells against infections. However, the important, tissue resident CD4 T cell population was not mentioned. They are found in tonsils, salivary glands, lung and intestine – all important sites for COVID-19 replication. These cells should be mentioned in the list of limitations.

Assumingly, the samples collected as part of the Convalescent Plasma Donor 69 Vaccine Study -VELVET study was done during the COVID-19 pandemia. The date of the collection duration should be mentioned.

It should be mentioned that the material was collected during a very chaotic time including the uncertainty of the choice of vaccines. This will be forgotten when this information of a primary virus infection is read later.

This will put the information in its (honest) context. Why hide this information?

Minor points

In the supplementary Table 1 - TCR matched epitopes, a number of epitopes against frequent viral infections are found. Interestingly, HSV1/2 epitopes a frequent latent virus were not found. Is there an explanation for this?

On the same Table: Please explain “vaccinated” as opposed to “one dose” “two dose”

Reviewer 4 Report

Comments and Suggestions for Authors

Mosavie et al. studied alterations in CD4+ T-cell subsets and molecular features of CD4+ naïve and central memory subsets between SARS-CoV-2 unvaccinated and vaccinated groups by using CyTOF. The study is an interesting approach with a novel system. But there are several issues.

First of all, figures (Figs. 1 to 4) are not readable even they are enlarged. They must be improved.

Authors used and compared three groups, unvaccinated group, group vaccinated with two doses of SARS-CoV-2 vaccine, and group vaccinated with one dose of SARS-CoV-2 vaccine. Conditions of the samples (time after vaccination, etc.) are critical, in order to characterize the T cells. Conditions of samples should be described in more detail to the extent possible. Otherwise, a detailed analysis is meaningless.

Round 2

Reviewer 1 Report

Comments and Suggestions for Authors

All my comments have been solved. 

Reviewer 4 Report

Comments and Suggestions for Authors

Unfortunately, the revised manuscript still has critical issues. First of all, Figures (Figs. 1 to 5) are still not readable even they are enlarged. Authors should select essential data for figures in manuscript and show other data as supplementary figures. Authors used and compared three groups, unvaccinated group, group vaccinated with two doses of SARS-CoV-2 vaccine, and group vaccinated with one dose of SARS-CoV-2 vaccine. Conditions of the samples (time after vaccination, age, etc.) are critical, in order to characterize the T cells. According to Table 3, ages of two dose group participants are higher compared with those of unvaccinated and one dose groups. The difference may affect the results.

Round 3

Reviewer 4 Report

Comments and Suggestions for Authors

The revised manuscript still has critical issues.

Figures are still not readable.
